



# Plant genotype controls wetland soil microbial functioning in response to sea-level rise

Hao Tang[1], Susanne Liebner[2,3], Svenja Reents[1], Stefanie Nolte[4,5], Kai Jensen[1], Fabian Horn[2] and Peter Mueller[1,6]

[1]Institute of Plant Science and Microbiology, Universität Hamburg, Hamburg, 22609, Germany
[2]GFZ German Research Centre for Geosciences, Geomicrobiology, Potsdam, 14469, Germany
[3]Institute of Biochemistry and Biology, University of Potsdam, Potsdam, 14469, Germany
[4]School of Environmental Sciences, University of East Anglia, Norwich, NR47TJ, UK
[5]Centre for Environment, Fisheries and Aquaculture Science, Pakefield Rd, Lowestoft, UK
[6]Smithsonian Environmental Research Center, Edgewater, MD 21037, United States

*Correspondence to*: Peter Mueller (muellerpe@si.edu; peter.mueller@uni-hamburg.de)

**Abstract.**

Climate change induced shifts in plant community composition affect the decomposition of soil organic matter via plant-microbe interactions, often with important consequences for ecosystem carbon and nutrient cycling. Given the high degree of

intraspecific trait variability in plants, it has been hypothesized that genetic shifts within species yield a similar potential to affect soil microbial functioning.

We examined if sea-level rise and plant genotype interact to affect soil microbial communities in an experimental coastal wetland system, using two known genotypes of the dominant salt-marsh grass *Elymus athericus* characterized by differences in their sensitivity to flooding stress – i.e. an adapted genotype from low-marsh environments and an unadapted genotype from

high-marsh environments. Plants were exposed to a large range of flooding frequencies in a factorial mesocosm experiment, and soil microbial-activity parameters (exo-enzyme activity and litter breakdown) and microbial community structure were assessed.

Plant genotype mediated the effect of flooding on soil microbial community structure and determined the presence of flooding effects on exo-enzyme activities and belowground litter breakdown. Larger variability in microbial community structure,

enzyme activities, and litter breakdown in soils planted with the unadapted plant genotype supported our general hypothesis that effects of climate change on soil microbial activity and community structure can depend on plant intraspecific adaptations. We conclude that adaptive genetic variation in plants can suppress or facilitate the effects of climate change on soil microbial communities. If this finding applies more generally to wetland ecosystems and beyond, it yields important implications for experimental climate change research and models of soil organic matter accumulation.



## 1 Introduction

Climate change strongly affects soil microbial decomposition, with important consequences for global carbon (C) and nutrient cycles (Davidson and Janssens, 2006; Dijkstra et al., 2010). Plant-microbe interactions in the rhizosphere are particularly susceptible to various climate change factors (Philippot et al., 2013; Pugnaire et al., 2019; Wieder, 2014). It is therefore crucial to not only study the direct effects of climate change on soil microbial communities and resulting changes in ecosystem
functioning, but also the plant-mediated, indirect effects (Bardgett et al., 2008; Van der Putten et al., 2013). Indeed, several case studies from a wide range of ecosystems demonstrated how changes in plant productivity and community composition control soil microbial functioning in response to climate change, often with marked effects on ecosystem C, greenhouse-gas, and nutrient dynamics (Fuchslueger et al., 2014; Mueller et al., 2020; Stagg et al., 2018; Ward et al., 2013).

Climate change does not only cause shifts in plant community composition, but also affects the genetic structure within plant populations (Bustos-Korts et al., 2018; Crutsinger et al., 2006; Jump and Peñuelas, 2005). Given the high degree of intraspecific trait variability in plants, it has been hypothesized that genetic shifts within plant populations yield the potential to induce important changes in soil microbial functioning (Fischer et al., 2014; terHorst and Zee, 2016; Van Nuland et al., 2016; Ware et al., 2019). This hypothesis is based on studies demonstrating differences in soil microbial community structure
or activity in soils of different plant genotypes (Madritch and Lindroth, 2011; Pérez-Izquierdo et al., 2019; Schweitzer et al., 2008; Seliskar et al., 2002; Zogg et al., 2018). Furthermore, genotype effects on soil C and nitrogen (N) stocks as well as N transformations have been observed to be variable across multiple common garden sites (Pregitzer et al., 2013). However, experimental evidence for interaction effects of plant intraspecific variability and climate change factors on soil microbial processes and C cycling is virtually absent.

Plant-mediated climate change effects on soil microbial functioning are expected to be particularly pronounced in wetlands, because here plants do not only control the microbial substrate (i.e. electron donor) supply, they also regulate the availability of electron acceptors by providing oxygen to an otherwise reducing rhizosphere (Kirwan and Megonigal, 2013; Wolf et al., 2007). At the same time, wetland soil microbial functioning plays a disproportionately large role in the global climate system
(Freeman et al., 2001; Megonigal et al., 2003). In recent years, climate change research in tidal wetlands and other so-called blue carbon ecosystems has gained increasing attention by the scientific community (Kirwan et al., 2013, 2014; Spivak et al., 2019). These ecosystems belong to the most effective long-term C sinks of the biosphere (Chmura et al., 2003; McLeod et al., 2011), but the impacts of accelerated rates of sea-level rise (SLR) destabilize tidal wetlands worldwide (Kirwan and Megonigal, 2013).

SLR affects the flooding frequency of tidal wetlands and represents the overriding climate change factor impacting tidal wetlands (Kirwan and Megonigal, 2013). Its effects on ecosystem functioning are largely plant-mediated and extremely





variable, ranging from strong positive effects on soil C sequestration to ecosystem destabilization and ultimately loss (Rogers
et al., 2019). SLR and the resulting flooding frequency alter plant primary production and microbial decomposition, the two
primary factors controlling C sequestration in coastal marine ecosystems (Kirwan and Megonigal, 2013). Primary production
often follows a unimodal (i.e. optimum) response to SLR, although interspecific variability is high (Kirwan and
Guntenspergen, 2012; Morris et al., 2013). The microbial decomposition response to SLR is less understood. A dominant
paradigm in wetland ecosystem ecology is that decomposition rates are inversely related to flooding. However, recent studies
demonstrated that the responses of decomposition and primary production to SLR are coupled (Janousek et al., 2017; Mueller
et al., 2016; Stagg et al., 2017). For instance, Mueller et al. (2016) demonstrated soil microbial activity is not directly affected
by SLR and its control on soil oxygen availability, but indirectly by the aboveground-biomass response to flooding frequency,
which determines the input of both oxygen and labile substrates to soil microbial communities.

Considering the low plant species diversity of many wetland types, such as salt marshes and ombrotrophic peatlands (Wanner
et al., 2014; Warner and Asada, 2006), and the strong control of plant processes on microbial C cycling in wetland soils, it is
possible that intraspecific variation and adaptive capacity functions as an important, but so far overlooked mediator of wetland-
climate feedbacks. Here, we study the interaction effect of flooding frequency and plant genotype on soil microbial community
structure and functioning, using the dominant tidal-wetland grass *Elymus athericus* as a model species (Bockelmann and
Neuhaus, 1999). Two genotypes of *Elymus athericus,* which differ in their adaptation to flooding frequency, have been
identified: a flooding-sensitive genotype from the high marsh (to simplify hereafter referred to as unadapted genotype) and a
less flooding-sensitive genotype from the low marsh (hereafter adapted genotype) (Bockelmann et al., 2003; Reents et al.,
2021). Given the overriding control of plant processes on microbial functioning in wetland soils, we hypothesize that flooding
effects on microbial decomposition and microbial community structure are strong in soils with the unadapted plant genotype,
but absent or buffered in soils of the adapted plant genotype (Figure 1).



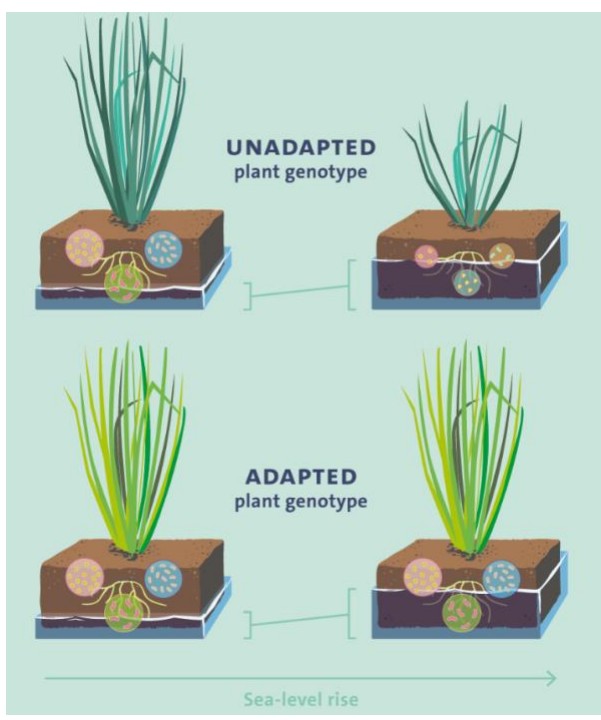


**Figure 1: Conceptual diagram illustrating the hypothesis that effects of a changing abiotic environment on soil microbial processes are mediated by the intraspecific adaptive variation of plants. We tested this general hypothesis in a tidal-wetland system and studied the interaction effect of plant genotype and flooding frequency (a master variable in tidal-wetland ecology that will increase with accelerated sea-level rise) on soil microbial functioning. Two genotypes of the dominant tidal-wetland grass *Elymus athericus* have**
**been identified, an unadapted plant genotype, found in high-marsh environments, and an adapted plant genotype, found in low-marsh environments. The adapted genotype shows no reduction of aboveground biomass even in response to extreme increases of flooding frequency (Reents et al., 2021). Given the overriding control of plant aboveground processes on microbial functioning in tidal wetland soils, we hypothesize that the adapted genotype buffers the response of the soil microbial community to increasing flooding frequency.**

## 95 2 Method

### 2.1 Experimental design

The experiment has been conducted from July to October 2017 (12 weeks) at the Institute of Plant Science and Microbiology (IPM), Universität Hamburg, Germany. We used platforms positioned at three elevations in a 12 m³ tidal tank to induce three flooding-frequency treatments capturing the full range of flooding frequencies of a typical NW European salt marsh: daily
(two floods every day, simulating pioneer-zone conditions), weekly (2 floods on one day per week, simulating low-marsh conditions), and monthly (2 floods on one day every two weeks, simulating high-marsh conditions). Similar experimental designs have previously been described as marsh organs (Mueller et al., 2016). Mesocosms (Ø = 15 cm; h = 17 cm) were filled with soils collected from the upper 25-cm soil layer of the high-marsh platform of a salt marsh near Sönke-Nissen-Koog, Germany (DE, 54°36'N, 8°49'E). The collected soils had low organic matter contents of 3 - 4%, low C:N ratios of 14-16, and
relatively high pH of 7.5 - 8.0, which are typical features of the minerogenic marshes of the European North Sea (Mueller et

al., 2019). Soils were sieved using a 1-cm mesh to remove roots, rhizomes, and other coarse materials, and homogenized before being transferred to the mesocosms. Mesocosms were planted with either adapted or unadapted genotypes of the grass *Elymus athericus*. The design included a total of 48 planted mesocosms (24 per genotype, 16 per flooding treatment) resulting from n = 8 unadapted versus n = 8 adapted genotypes per flooding treatment. We additionally added four unplanted mesocosms to

each flooding treatment to understand the direct (i.e. plant-independent) effect of flooding on soil microbial communities and thus gain more mechanistic insight into potential genotype effects (n = 4).

Plants were collected in April 2015 from *Elymus athericus* stands on the island Schiermonnikoog, the Netherlands, that have previously been demonstrated to be dominated by genetically distinct populations of *Elymus*, i.e. unadapted and adapted

genotypes (Bockelmann et al., 2003; Reents et al., 2021). The plants were transferred to pots and kept in a common garden at Universität Hamburg for 24 months before the experiment commenced. Clonal plant growth led to the emergence of new individuals during this period, which were used for the experiment. New individuals of unadapted and adapted genotypes were still phenotypically distinct after 24 months under identical environmental conditions. Each mesocosm received one plant of similar size (compare Reents et al., 2021 for details).

**2.2 Soil sampling and processing**

Soil sampling took place in October 2017 after 12 weeks of exposure to different flooding treatments and plant genotypes. Plant biomass and litter were removed before sampling. From each mesocosm, one soil sample was taken as a 5-cm diameter and 5-cm deep core using a volumetric steel ring. Sub-samples of 20 g were homogenized and stored frozen until used for microbial enzyme assays and DNA extraction. The residual sample was passed through a 2.5-mm sieve, air-dried at 65 °C to

constant weight, and used to determine dry mass and other soil properties.

**2.3 Microbial exo-enzyme activity and belowground litter decomposition**

Potential exo-enzyme activity (EEA) of ß-glucosidase, cellobiosidase, leucine-aminopeptidase, and chitinase was determined in fluorometric assays following Mueller et al. (2017). Briefly, 1:20 soil slurries were produced using 50 mmol/L bicarbonate buffer (pH = 8) (Sinsabaugh et al., 2003). 96-well-plate assays were conducted to measure potential EEA. Plates were

incubated in the dark at 20 °C for 16 h and read on a Multi-Detection Microplate Reader (Bio-Tek Synergy HT, Winooski, USA). The emission and excitation wavelengths were set at 460 nm and 365 nm, respectively. The four enzymes assayed are commonly used as proxies for microbial C- and N-acquisition activities that reflect the microbial C and N demand (Sinsabaugh et al., 2009; Sinsabaugh et al., 2008).

We assessed the decomposition of standardized plant litter in the rhizosphere to evaluate if genotype effects on soil microbial exo-enzyme activity translate into altered organic matter turnover and thus into ecosystem functioning (Ochoa-Hueso et al., 2020). The decomposition rate constant (*k*) and stabilization factor (*S*) were assessed following the Tea Bag Index (TBI)





protocol (Keuskamp et al., 2013). The TBI is a standardized litter-decay assay using commercially available tea materials as standardized plant litter. The TBI has widely been applied to characterize and compare decomposition dynamics within and

across ecosystems (Keuskamp et al., 2013; Mueller et al., 2018; Ochoa-Hueso et al., 2020). The advantages and limitations of the TBI and other standardized decomposition assays, such as cotton- and cellulose-strip assays, have been extensively discussed elsewhere (Clark, 1970; Mueller et al., 2018; Ochoa-Hueso et al., 2020; Risch et al., 2007). Each pot received, two polypropylene tea bags (55 mm x 50 mm), one containing green tea (EAN: 8 714100 770542; Lipton, Unilever), and one containing rooibos (EAN: 8 722700 188438; Lipton, Unilever). Bags were deployed in 5 cm soil depth. The initial weight of

the contents was determined by subtracting the mean content weight of 5 empty bags (Green tea: $1.69 \pm 0.005$ g; Rooibos tea: $1.79 \pm 0.009$ g). Bags were retrieved after an incubation period of 90 days, carefully separated from roots and soil, dried for 48 h at 70 °C, and weighed. The TBI parameters $k$ and $S$ were calculated following the tidal-wetland-adapted TBI protocol (Mueller et al., 2018).

## 2.4 Microbial community structure - Illumina sequencing

Soil DNA was extracted from n = 3 randomly chosen mesocosms per treatment combination using the PowerSoil DNA extraction kit (Quiagen). From each mesocosm, two samples (technical replicates) were taken to assess within-mesocosm variability. DNA quality and yield were assessed using a fluorometer (Qubit 2.0, Thermo Fisher Scientific). PCR amplification of the prokaryotic 16S rRNA gene region was conducted using the barcoded primers 515F (5'-GTGCCAGCMGCCGCGGTAA-3') and 806R (5'-GGACTACHVGGGTWTCTAAT-3') (Caporaso et al., 2010). The PCR

protocol (PCR mix and cycling conditions) followed Meier et al., (2019). PCR products were purified using the Agencourt AMPure XP– PCR purification kit (Beckman Coulter, Inc.), and were pooled into a single sequencing library at equimolar concentrations (20 ng DNA per sample). Sequencing was conducted by Eurofins Scientific (Konstanz, Germany) using an Illumina HiSeq platform and Miseq v3 kits (2 x 300 bp). Sequence analysis and bioinformatics followed Holm et al., (2020). Briefly, the library was demultiplexed using Cutadapt (Martin, 2011), and samples were error-corrected using the DADA2

pipeline (Callahan et al., 2016). Paired-end reads were merged, and low-quality sequences and chimeras were removed. Amplicon sequence variants (ASV) were assigned to the SILVA database (version 132) (Quast et al., 2013) applying vsearch (Rognes et al., 2016) as implemented in the QIIME2 framework (Bolyen et al., 2019). Taxonomic assignment of sequences was based on a 99% similarity threshold. Raw sequencing data is available at the European Nucleotide Archive (ENA) under BioProject accession number PRJEB38150 and sample accession numbers ERS4541081-ERS4541134.

## 2.5 Statistical analyses

We used two-way ANOVA or two-way PERMANOVA to analyze the data of our two-factorial design (2 genotypes x 3 flooding frequencies). Specifically, two-way ANOVA was conducted to test for effects of flooding frequency, plant genotype, and their interaction on EEAs, $k$, and $S$. Data on EEAs, $k$, and $S$ are presented both as absolute values and in relation to the mean of the unplanted mesocosms of each flooding treatment (i.e. percentage change versus unplanted conditions = ΔEEA,





Δ*k*, Δ*S*). This was done to explore potential differences in magnitude and direction of plant effects between genotypes. Two-way PERMANOVA, based on Bray-Curtis dissimilarities, was used to test for effects of flooding frequency and genotype on microbial community composition. Data of technical replicates were averaged for two-way PERMANOVA. Data were visualized using NMDS displaying all technical replicates. In addition to these two-factorial tests, we conducted a paired t-test to compare effect sizes of the flooding treatment on EEAs between genotypes, and Pearson correlation and Canonical

Correspondence Analysis (CCA) to explore the relationships between soil microbial parameters and plant biomass parameters (taken from Reents et al., 2021). The analysis of flooding effects on soil microbial parameters in the absence of plants can provide additional mechanistic insight but was not the primary objective of our study. To facilitate a clearer presentation of genotype and genotype-flooding-interaction effects, flooding effects in the absence of plants were analyzed separately (i.e. not as part of our 2-factorial design) using one-way ANOVA or one-way PERMANOVA.

**3 Results**

**3.1 Soil microbial enzyme activity and litter decomposition**

Enzyme activities were only affected by flooding frequency in soils planted with the unadapted genotype, whereas none of the four EEAs were affected in soils planted with the adapted genotype (Table 1). In soils with the unadapted genotype, all four EEAs showed a unimodal response to flooding: They were always highest at the intermediate (i.e. weekly) flooding frequency

and always lowest at the highest (i.e. daily) flooding frequency, whereas no consistent pattern was found in soils of the adapted genotype (Table 1, Figure 2). Overall, the effect size of flooding frequency (i.e. the difference between highest and lowest mean activity of the three flooding treatments) was 1.7 - 4.7 times greater in the unadapted vs. adapted genotype (Figure 3).

C-acquisition enzymes (ß-glucosidase and cellobiosidase, *sensu* Sinsabaugh et al., 2009) showed different responses than N-

acquisition enzymes (leucine-aminopeptidase and chitinase, *sensu* Sinsabaugh et al., 2009). The activity of C-acquisition enzymes was not affected by flooding frequency, genotype, and their interaction (Figure 2A, Table 1), whereas N-acquisition enzymes were significantly reduced by the highest flooding frequency (Figure 2B, Table 1). The reduction of N-acquisition activities by increasing flooding frequency was only observed in the unadapted genotype, whereas activities remained unchanged throughout flooding treatments in the adapted genotype (Figure 2B).


Analyzing the EEA data in relation to the activity under unplanted conditions reveals contrasting plant effects between genotypes (Figure 2). Specifically, at our highest flooding frequency, the activity change in relation to the unplanted condition was negative in the unadapted genotype but positive in the adapted genotype (Figure 2). This contrasting pattern in the direction of plant effects was generally found for all enzymes assayed, but it was significant in the N enzymes only (Figure 2). The

absolute values of enzyme activities under unplanted conditions are presented in the Supplementary Material. None of the four





enzymes assayed showed a significant response to changes in flooding frequency under unplanted conditions (Figure S1, Table S1).

**Table 1: Exo-enzyme activities (nmol·g DW⁻¹·h⁻¹) of ß-glucosidase (GLU), cellobiosidase (CEB), chitinase (CHI), and leucine-aminopeptidase (LAP) as well as the litter-breakdown parameters *k* (decomposition rate constant) and *S* (stabilization factor) in soils planted with unadapted and adapted plant genotypes of *Elymus athericus* exposed to three different flooding frequencies (monthly, weekly, and daily). Values are means and SE (n = 8). Values not connected by the same letter within one column are significantly different at p ≤ 0.05 based on Tukey's HSD tests. Corresponding two-way ANOVA results are included below (p-values highlighted in bold font at p ≤ 0.05).**

| Genotype | Flooding | GLU | | CEB | | CHI | | LAP | | $k$ | | $S$ | |
|---|---|---|---|---|---|---|---|---|---|---|---|---|---|
| Unadapted | Monthly | 32.79 ± 7.67 a | | 14.85 ± 3.70 a | | 13.00 ± 0.88 ab | | 58.19 ± 5.72 a | | 0.008 ± 0.000 b | | 0.13 ± 0.02 b | |
| | Weekly | 45.05 ± 11.56 a | | 17.32 ± 4.70 a | | 16.74 ± 2.42 a | | 60.40 ± 7.07 a | | 0.015 ± 0.003 a | | 0.25 ± 0.02 a | |
| | Daily | 27.95 ± 5.11 a | | 9.23 ± 1.84 a | | 10.67 ± 1.22 b | | 38.64 ± 1.19 b | | 0.010 ± 0.001 ab | | 0.21 ± 0.02 a | |
| Adapted | Monthly | 39.24 ± 4.83 a | | 13.91 ± 1.18 a | | 13.72 ± 0.60 ab | | 69.49 ± 2.78 a | | 0.009 ± 0.002 ab | | 0.24 ± 0.03 a | |
| | Weekly | 40.99 ± 6.65 a | | 16.87 ± 2.80 a | | 12.86 ± 0.72 ab | | 56.86 ± 2.60 ab | | 0.011 ± 0.001 ab | | 0.23 ± 0.02 a | |
| | Daily | 37.35 ± 7.38 a | | 13.81 ± 3.59 a | | 14.33 ± 0.84 ab | | 62.39 ± 0.60 a | | 0.011 ± 0.001 ab | | 0.12 ± 0.02 b | |
| *Two-way ANOVA results* | | F | p | F | p | F | p | F | p | F | p | F | p |
| Flooding | | 0.9 | 0.385 | 1.5 | 0.232 | 1.6 | 0.210 | 4.6 | **0.016** | 3.9 | **0.029** | 5.9 | **0.006** |
| Genotype | | 0.4 | 0.528 | 0.2 | 0.685 | 0.0 | 0.875 | 8.6 | **0.013** | 0.7 | 0.412 | 0.0 | 0.928 |
| Flooding x Genotype | | 0.4 | 0.647 | 0.5 | 0.640 | 4.3 | **0.020** | 4.8 | **0.013** | 1.8 | 0.180 | 11.2 | **0.000** |

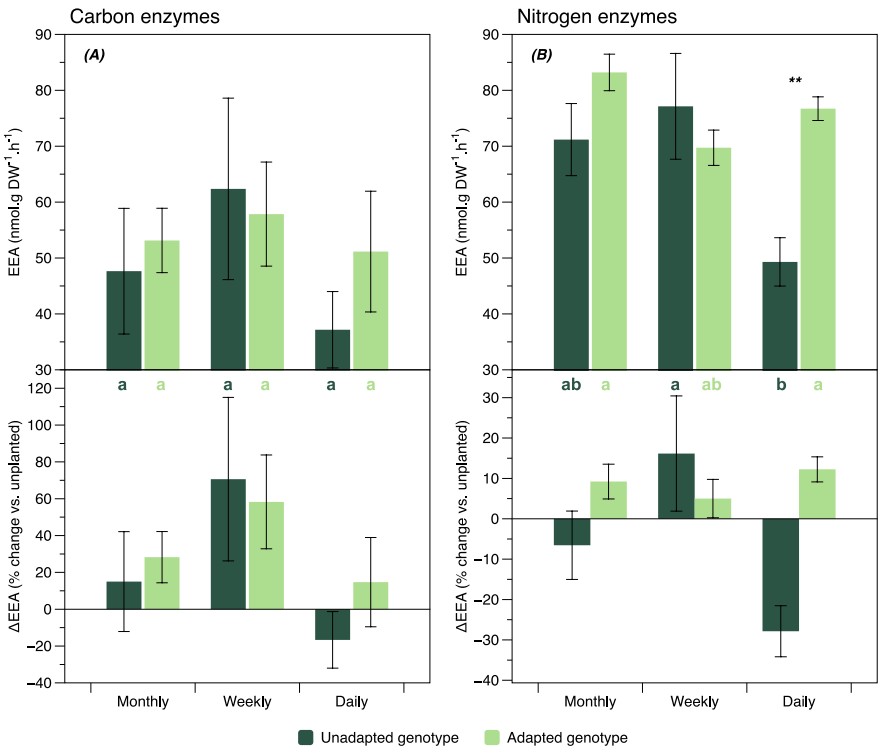

**Figure 2: Exo-enzyme activity (EEA) of C-acquisition enzymes (A; sum of ß-glucosidase and cellobiosidase) and N-acquisition enzymes (B; sum of leucine-aminopeptidase and chitinase) in soils planted with unadapted and adapted plant genotypes of *Elymus athericus* exposed to three different flooding frequencies (monthly, weekly, and daily). Upper panels show absolute values, and lower panels show activities in relation to the unplanted control (i.e. percentage change vs. the mean value of n = 4 unplanted mesocosms per flooding treatment). Values are means and SE (n = 8). Asterisks denote significant genotypic differences within the same flooding treatment (\* = p ≤ 0.05; \*\* p ≤ 0.01). Bars not labeled by the same letter are significantly different at p ≤ 0.05. All statistical results refer to the absolute enzyme data shown in the upper panels and are based on Tukey's HSD tests following two-way ANOVA.**

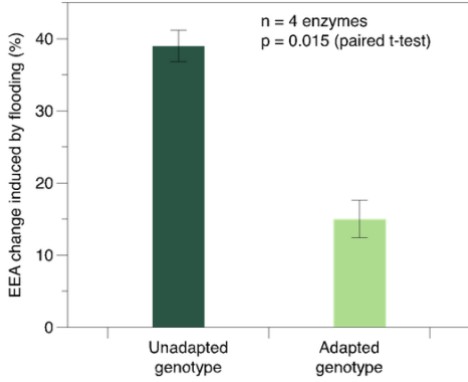

**Figure 3: Maximum change in exo-enzyme activity (EEA) induced by the flooding treatment, i.e. the difference between highest and lowest mean activity of the three flooding treatments, in soils planted with unadapted and adapted plant genotypes of *Elymus athericus*. Values are means and SE (n = 4 enzymes; compare Table 1).**

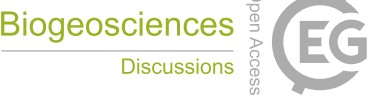

The initial belowground litter decomposition rate, $k$ (*sensu* Keuskamp et al., 2013), was significantly affected by flooding
frequency. However, based on pairwise comparisons, this effect was only significant in the unadapted plant genotype (Figure
4A), reflecting the greater flooding sensitivity of the soil microbial community that has also been observed in EEAs. A strong
interaction effect of flooding frequency and genotype was detected on the litter stabilization factor, $S$ (Keuskamp et al., 2013)
(Figure 4B). At the highest flooding frequency, $S$ was markedly lower in the rhizosphere of the adapted versus unadapted
genotype, whereas the reversed pattern was found at our lowest (i.e. monthly) flooding frequency (Figure 4B).

Significant relationships between plant biomass parameters (taken from Reents et al., 2021), soil EEAs, and litter-breakdown
parameters have been observed (Table 2). C enzymes were not significantly related to any plant biomass parameter, reflecting
the missing plant genotype effect on microbial C-enzyme activities, whereas N-enzyme activities were significantly positively
related to plant aboveground biomass (Table 2). Relationships between plant biomass parameters and litter-breakdown
parameters ($k$, $S$) were only significant when controlling for direct (i.e. plant-independent) flooding effects (Table 2).
Specifically, while $\Delta S$ was significantly related to both above- and belowground biomass, $\Delta k$ was only related to belowground
biomass (Table 2). $k$ and $\Delta k$ were most strongly related to C-enzyme activities, whereas $S$ was not significantly related to
EEAs, and $\Delta S$ was significantly related to N-enzyme activities (Table S2).

**Table 2: Correlations between plant biomass parameters and soil microbial activity parameters. Shown are Pearson correlation**
**coefficients (r). Significant (p ≤ 0.05) correlations are highlighted in bold font.**

| | Aboveground | Belowground | Total biomass |
|---|---|---|---|
| C activity | 0.03 | -0.02 | 0.01 |
| N activity | **0.41** | 0.23 | **0.36** |
| ΔC activity | 0.06 | -0.01 | 0.02 |
| ΔN activity | **0.37** | 0.09 | 0.26 |
| Decomp. rate *(k)* | -0.11 | -0.06 | -0.10 |
| Stabilization *(S)* | -0.05 | -0.15 | -0.12 |
| Δk | -0.26 | **-0.36** | **-0.35** |
| ΔS | **-0.41** | **-0.45** | **-0.49** |

*Notes:* C activity = sum of C-acquisition enzyme activities (ß-glucosidase + cellobiosidase); N activity = sum of N-acquisition enzyme
activities (aminopeptidase + chitinase); Decomp. rate ($k$) = decomposition rate constant (*sensu* Keuskamp et al. 2013); Stabilization ($S$) =
stabilization factor (*sensu* Keuskamp et al. 2013); Δ = activity values in relation to the unplanted control (i.e. percentage change of planted
vs. unplanted mesocosms) reflecting plant effects independent of direct (i.e. non-plant mediated) flooding effects.

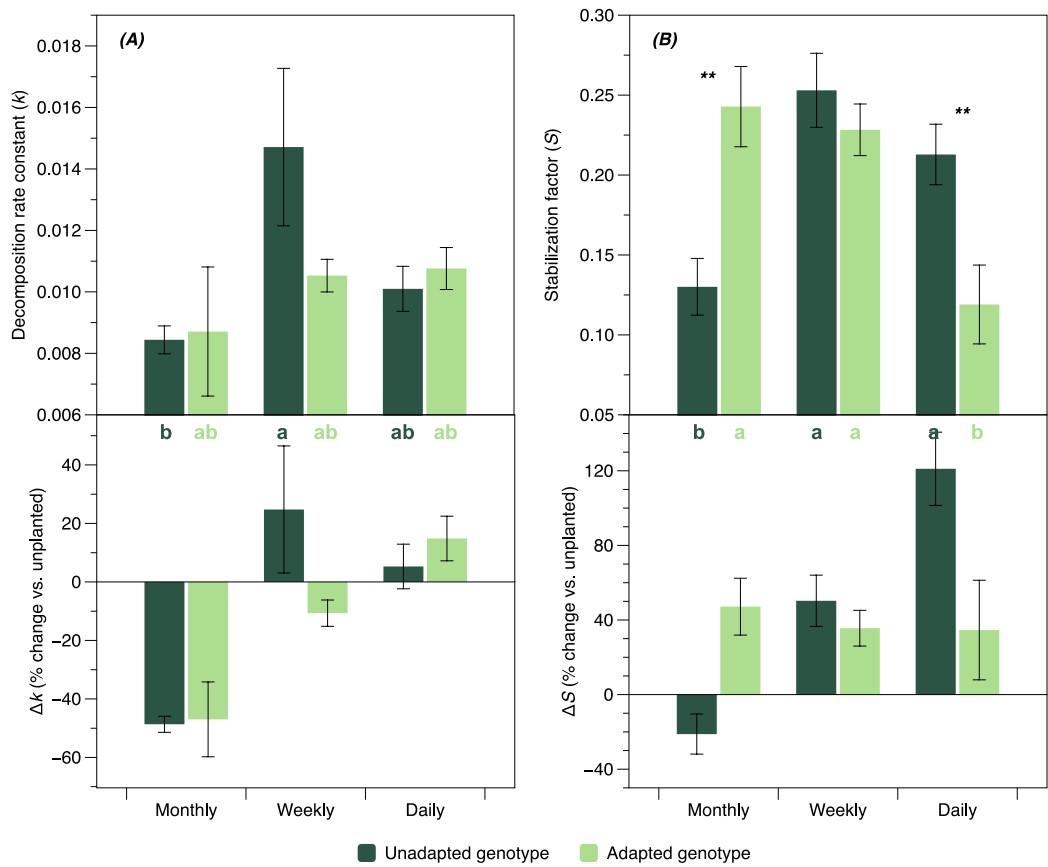


**Figure 4:** Initial decomposition rate constant (*k*) and stabilization factor (*S*) (*sensu* Keuskamp et al., (2013)) in soils planted with unadapted and adapted plant genotypes of *Elymus athericus* exposed to three different flooding frequencies (monthly, weekly, and daily). Upper panels show absolute values, and lower panels show activities in relation to the unplanted control (i.e. percentage change vs. the mean value of n = 4 unplanted mesocosms per flooding treatment). Values are means and SE (n = 8). Asterisks denote significant genotypic differences within the same flooding treatment (* = p ≤ 0.05; ** p ≤ 0.01). Bars not labelled by the same letter are significantly different at p ≤ 0.05. All statistical results refer to the absolute enzyme data shown in the upper panels and are based on Tukey's HSD tests following two-way ANOVA.

### 3.2 Soil microbial community structure

Flooding frequency (2-way PERMANOVA, F = 2.33, p ≤ 0.001) and plant genotype (F = 2.09, p ≤ 0.001) significantly affected the microbial community structure (Figure 5). In accordance with the findings on EEAs, genotype effects were most pronounced at the highest (i.e. daily) flooding frequency treatment (Figure 5). By contrast, differences between genotypes were absent at the lowest, i.e. monthly flooding frequency, suggesting an interaction of genotype and flooding frequency on soil microbial community structure (Figure 5), which was, however, not statistically significant based on two-way

PERMANOVA (F = 1.08; p ≥ 0.1). Overall, variability in microbial community structure across flooding treatments was greater in the unadapted vs. adapted plant genotype (Figures 5), reflecting the findings on EEAs and *k*. Canonical Correspondence Analysis (Figure S2) indicates that soil microbial community is significantly related to plant biomass parameters as well as to microbial C and N demands. Aboveground biomass exerted the strongest effect on the community structure (Figure S2). The overview of the most abundant prokaryotic taxa was shown in Figure S3. However, owing to the

artificial nature of the simulated tidal-wetland system used in our study, it was not our objective to identify and discuss the specific microbial taxa affected by genotype or flooding treatments.

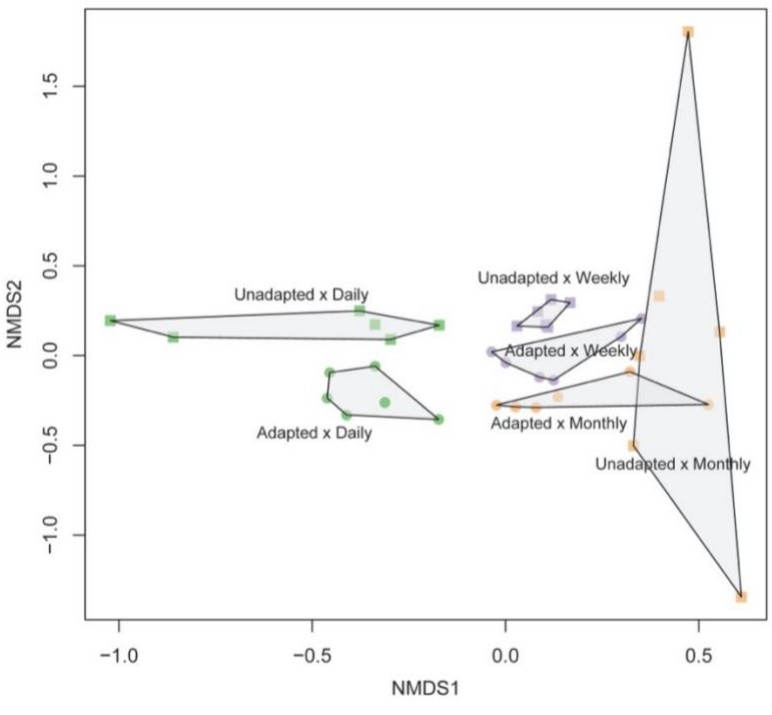

**Figure 5: NMDS plot showing prokaryotic (bacterial + archaeal) community composition in soils planted with unadapted and**
**adapted plant genotypes of *Elymus athericus* exposed to three different flooding frequencies (monthly, weekly, and daily). Plot shows all technical replicates (i.e. two samples from n = 3 mesocosms). For PERMANOVA analyses data from technical replicates were averaged.**

## 4 Discussion

The present study provides experimental evidence of genotype-environment interaction effects on soil microbial enzyme
activity (Figures 2, 3) and belowground litter breakdown (Figure 4), two key processes controlling C and nutrient cycling in ecosystems. Plant genotype determined the presence or absence of climate change effects (here increasing flooding frequency) on microbial enzyme activities and litter breakdown. This result yields important implications for our understanding of soil-





climate feedbacks, because it shows that plant-genotype controls can mask or enhance the effects of changing abiotic conditions on soil microbial processes. Our data furthermore suggest genotype-environment interaction effects on the soil

microbial community structure (Figure 5). This finding is in agreement with a recent observational study suggesting that climate-driven reduction of genetic variation in *Populus angustifolia* phenology affects soil fungi-to-bacteria ratios (Ware et al., 2019), and a laboratory experiment demonstrating interaction effects of drought and rapid evolution in *Brassica rapa* on soil microbial community structure (terHorst et al., 2014). Overall, larger variability in microbial community structure (Figure 5), enzyme activities (Figure 2), and litter decomposition (Figure 4 A) in soils planted with the unadapted plant genotype

support our central hypothesis that effects of climate change on soil microbial activity and community structure depend on plant intraspecific adaptations.

### 4.1 Genotype aboveground-biomass response controls flooding effects on soil microbial functioning

The majority of studies on genotype-environment interactions is concerned with plant responses to temperature or latitudinal climate gradients in terrestrial ecosystems (Bauerle et al., 2007; Curasi et al., 2019; Taylor et al., 2019; Walker et al., 2019;

Ware et al., 2019). Here, we manipulated flooding frequency to simulate SLR, the overriding climate change factor in coastal ecosystems, such as tidal wetlands. The effects of SLR on soil microbial activity can be tightly controlled by the plant response to changes in flooding frequency, as demonstrated by recent studies showing strong positive correlations between aboveground biomass and soil litter decomposition (Janousek et al., 2017), cellulose decomposition (i.e. tensile strength loss; Jones et al., (2018)), or recalcitrant soil organic matter decomposition (Mueller et al., 2016). The importance of plant processes in

controlling soil microbial functioning in response to changing flooding frequency is reflected in the findings of the present study: In the absence of plants, flooding frequency neither affected soil microbial enzyme activities nor the soil microbial community structure (Figure S1 and Figure S4). In the presence of plants, however, flooding frequency, genotype, and genotype-induced variability in plant biomass exerted significant effects on soil microbial activity and community structure. Most notably, microbial enzyme activities only responded to changes in flooding frequency when aboveground biomass

responded. Aboveground and belowground biomass across flooding treatments was unchanged in the adapted genotype, whereas the unadapted genotype showed a strong reduction of aboveground biomass at our highest flooding treatment (Reents et al., 2021). Consequently, only the flooding-sensitive unadapted genotype showed changes in soil microbial activity, whereas the adapted genotype was able to maintain microbial enzyme activities at a constant level over the entire flooding gradient (Table 1; Figure 2).


In support of the notion that the soil microbial activity response to increasing flooding frequency follows the response of plant aboveground processes, we found a significant relationship between aboveground biomass and microbial N-acquisition activity (aminopeptidase + chitinase activity, *sensu* Sinsabaugh et al., 2009, 2008) across all flooding treatments (r = 0.41; p ≤ 0.01, Table 2) and to an even larger degree within the daily flooding treatment (r = 0.63; p = 0.01), where effects on aboveground

biomass and N-acquisition activity existed (Table 2 and Figure 2; (Reents et al., 2021)). Soil enzyme activity is tightly



controlled by the balance of nutrient supply and demand (Sinsabaugh et al., 2008, 2012). It is therefore possible that the maintenance of N-rich aboveground plant biomass increased the soil microbial N demand and thus stimulated the mineralization of N from soil organic matter, a mechanism that has been discussed in the context of rhizosphere priming effects (Kuzyakov, 2002).

### 4.2 Genotype-environment interactions control belowground litter breakdown

To evaluate if genotype effects on soil microbial communities translate into altered organic matter turnover and thus ecosystem functioning, we assessed the decomposition of standardized plant litter in the rhizosphere. The parameters $S$ and $k$ describe the initial transformation process of biomass to soil organic matter, which is a key component of many tidal-wetland resilience models that have highlighted the critical role of the organic contribution to wetland elevation gain (Schile et al., 2014; Swanson et al., 2014). Although actual rates of $S$ and $k$ cannot be inferred from TBI assays using standardized litter, the approach has proven a powerful tool to characterize the potential of the soil environment to transform organic matter inputs (Keuskamp et al., 2013; Mueller et al., 2018; Ochoa-Hueso et al., 2020). Effect sizes of the flooding treatment on $S$ and $k$ observed here are similar in range to those reported from field sites (Tang et al., 2020), and genotype effect sizes were surprisingly large. Specifically, differences in $S$ between genotypes within flooding treatments corresponded to c. 20% of the total range reported for tidal wetlands worldwide (Mueller et al., 2018). This result illustrates that the effects of plant genotype and genotype-climate change interactions on the C balance of tidal wetlands are not restricted to shifts in plant performance and primary production (Reents et al., 2021), but also concern parts of the soil C turnover.

Although significant correlations between microbial-activity and plant-biomass parameters were found, these are insufficient to clearly identify functional-trait differences between genotypes that control soil microbial functioning. Plants can control soil microbial activity and ultimately the decomposition of different soil organic matter pools via at least three non-exclusive mechanisms: (1) supplying oxygen to an otherwise anoxic soil system via root oxygen loss (Wolf et al., 2007); (2) competing with microbial communities for nutrients (Kuzyakov and Xu, 2013); (3) supplying of labile microbial substrates via rhizodeposition or root exudation (Jones et al., 2004; Kuzyakov, 2002). Root oxygen loss (mechanism 1) is only relevant in oxygen-deficient soils, but strong genotype effects on belowground litter decomposition were also present in our well-aerated monthly-flooding treatment (Figure 4B). Therefore, root oxygen loss is unlikely to represent the primary and sole driver of the observed genotype effects. Differences in nutrient demand between genotypes (mechanism 2) are supported by the clear differences in aboveground biomass production (Reents et al., 2021) and soil microbial N-acquisition activities (Figure 2). However, these differences in biomass production and microbial N-acquisition were also restricted to our highest flooding frequency and cannot explain the changes in belowground litter decomposition we observed under lower flooding frequencies. We therefore hypothesize that genotypic differences in root exudation patterns (mechanism 3) could have played an important role in the studied system. Root exudates are a key component of the plant control on soil decomposition processes in terrestrial soils, and their quantity and quality are not necessarily related to plant biomass parameters (Henneron et al., 2020; Jones et al.,



**Biogeosciences** Open Access

Discussions

EGU

2004; Koelbener et al., 2010). Furthermore, differences in root-exudation patterns between genotypes are known to alter microbial community structures in terrestrial ecosystems (Micallef et al., 2009). For wetlands, however, the current understanding of root-exudate effects on soil decomposition dynamics is insufficient to explore this hypothesis more thoroughly without additional research (Dinter et al., 2019; Mueller et al., 2016). Taken together, our findings highlight the need for further investigations into rhizosphere-trait variability, plant-soil interactions, and the mechanisms of rhizosphere priming effects in wetland ecosystems.

**5 Conclusions**

Larger variability in microbial community structure, enzyme activities, and litter decomposition in soils planted with the unadapted plant genotype support our general hypothesis that effects of changing abiotic conditions on soil microbial activity and community structure depend on plant intraspecific adaptations. Our findings therefore suggest that intraspecific adaptive variation in wetland plants could represent an important factor determining the response of soil microbial communities and

soil C turnover to climate change. If our findings apply more generally to wetland ecosystems, and potentially beyond, they could yield important implications for experimental climate change research and models of soil C accumulation, because they show that plant-genotype controls can mask or enhance the effects of changing abiotic conditions on soil microbial processes. Future research will need to put more emphasis on the intraspecific variability in plant functional traits as well as climate-change driven intraspecific shifts in wetland plant communities, neither of which were part of the present investigation.

**Data availability**

All data presented in this paper are available upon reasonable request.

Raw sequencing data is available at the European Nucleotide Archive (ENA) under BioProject accession number PRJEB38150 and sample accession numbers ERS4541081-ERS4541134.

**Author contribution**

HT, SR, SN, KJ, and PM designed and set up the experiment. HT conducted enzyme and decomposition assays, analyzed the resulting data. SL, PM, and FH planned the molecular microbial work. PM conducted the molecular microbial lab work. FH carried out the bioinformatics and analyzed the molecular data. HT and PM wrote the original draft with input from all co-authors.



**Competing interests**

The authors declare that they have no conflict of interest.

**Acknowledgements**

We thank Chris Smit and his colleagues from Groningen University for the provision of plants, Max Beiße, Marion Klötzl, and Maren Winnacker at Hamburg University for assisting the experimental phase, and Anke Saborowski at GFZ for her assistance with lab work. Hao Tang acknowledges financial support from the China Scholarship Council. Peter Mueller is supported by the DAAD (German academic exchange service) PRIME fellowship program funded through the German Federal Ministry of Education and Research (BMBF).

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
