# Peer review of "Plant genotype controls wetland soil microbial functioning in response to sea-level rise"

_Biogeosciences, 2021_

## Author Comment (AC1)

We would like to thank both reviewers for their time and constructive comments. Below we respond to each comment separately (in blue font) referring to the line numbers of the original submission.

**Anonymous Referee #1**

General Comments:  The paper by Tang et al. uses an experimental marsh organ set up to examine the interaction of plant genotype (those adapted to flooding vs those unadapted) to flooding duration (daily, weekly, monthly). The authors hypothesized that flooding effects on microbial enzyme activity, decomposition, and community structure will be stronger in soils where the plant is not adapted to flooding than in systems where the plant is adapted to flooding. In general the paper is well written (though personally, I am not a fan of the extensive use of passive voice), the data are clear and well-presented and the results make an important contribution to the literature. My primary concern with the results, as presented, is in the lack of description for tests of assumptions for the statistical analyses.  No data are presented on whether tests of assumptions were performed prior to ANOVAs, the the enzyme activity and decomposition variables, nor for the PERMANOVA.  in fact, the authors highlight the differences in the variability of the microbial community as a function of genotype, which technically violates the assumption of equal variances that underlies PERMANOVA analyses. The conclusions of the paper would be stronger if the authors documented the results for the tests of assumptions (e.g. levene's test, betadisper, etc.) along with their statistical analyses. Lastly, both in the abstract and the discussion, the authors continually refer to the effects of climate change on microbial activity/structure. I would recommend that the authors use caution with this broader construct. Their work was very specifically about flooding, not the myriad other effects of climate change. The authors argue that flooding/sea level rise is the most important climate change impact in coastal systems, which may be true, but this does not change the fact that their experiment was not a multifaceted climate change experiment, it was a single environmental factor x genotype experiment. Therefore conclusions such as " adaptive genetic variation in plants can suppress or facilitate the effects of climate change on soil microbial communities" over-states what can be concluded from this work. I would recommend that the authors keep statements regarding the conclusions of this work firmly grounded in the effects of flooding, rather than the effects of climate change more broadly

We greatly appreciate these helpful comments.
(1) The statistical approach and experimental design chosen for our study is determined by its hypotheses, which require testing for the interaction effect of genotype and flooding treatment (i.e. GxE interaction). Therefore, 2-factorial PERMANOVA and ANOVA designs were used. Both statistical tests (i.e. PERMANOVA and ANOVA) are very robust to heterogeneity of variance for experiments with balanced designs (McGuinnes 2002; Anderson 2017), which is the case in the present study (n = 8 for all groups in ANOVA tests; n = 3 for all groups in PERMANOVA; full-factorial design). In the revised version of our ms, we will include this more detailed justification for the chosen statistical approaches.

For the revised version of the ms, we will also report on the results of beta dispersion tests, along with PERMANOVA. These indicate no significant variance heterogeneity.

Following the recommendations by McGuinnes (2002), we will assess the homogeneity assumption for ANOVAs using Cochrane's C test, which tests for the presence of a single large variance and is less sensitive to heterogeneity caused by small variances. Based on this assessment, only data on the decomposition rate ($k$) showed a potentially problematic degree of variance heterogeneity, which remained even after log-transformation. Because the decomposition rate *(k)* was not affected by genotype or its interaction with flooding, it did not play an important role in the data interpretation, discussion and conclusions of our work. For the revised version of the ms, we will use the nonparametric Krukal-Wallis test to illustrate the significant effect of flooding on this parameter. Importantly, a problematic degree of variance heterogeneity was not observed in any of the microbial activity parameters that were significantly affected by genotype or genotype X flooding interactions.

References:
McGuinness, K. A. (2002). Of rowing boats, ocean liners and tests of the ANOVA homogeneity of variance assumption. *Austral Ecology, 27*(6), 681-688.
Anderson, M. J. (2017). Permutational multivariate analysis of variance (PERMANOVA). *Wiley statsref: statistics reference online*, 1-15. https://doi.org/10.1002/9781118445112.stat07841

(2) In accordance with the reviewer's comments, we will change all statements in the abstract that expand our conclusions to climate change effects in general, and keep them specific to flooding or sea-level-rise effects. In the discussion, we will keep the comparisons to studies that looked at the effects of other climate change factors mediated through plant intraspecific variability. However, these sections will be carefully rephrased in order to avoid over-interpretation of our data.

Specific comments:

Line 49: this seems like an over-statement. There are several examples of work that has been done on linking plant infraspecific variability to C cycling, some of which was published by these authors (e.g. Mueller et al.).  I also find the "has not been done yet" argument to be less convincing than stating why it is important that the work be done. Perhaps switch this statement around to make it more clear why this study is needed and what knowledge gap it will fill rather than suggesting that there has been no work done on the topic.

We agree with the reviewer that there is a growing number of studies reporting on links between plant intraspecific variation and carbon cycling. However, we tried to specifically point to the lack of experimental proof for plant genotype x environment interactions effects on soil microbial C cycling via plant-soil interactions. Our statement in line 49  was not specific enough in the original submission. For the revised version, we will specify as follows: *"However, experimental evidence for interaction effects of plant intraspecific variability and climate change factors on soil microbial C cycling that operate through plant-soil interactions are virtually absent."*

Fig. 1: there is a decent amount of redundancy between the figure caption and the text, so if you are short on space you could edit this caption down.

This is a valid concern. We are aware of the redundancy between figure caption and text. We added detailed information to the caption, because we are convinced it will help readers

to quickly grasp the key points of our study without reading the entire manuscript. We are happy to streamline the information given in the caption but leave this decision to the editor.

Section 2.5: were tests of the assumptions underscoring these stats performed? These tests should be described in the methods and the results should be outlined in the results section.

We will elaborate on this in the methods section of the ms as follows:

*"We used two-way ANOVA or two-way PERMANOVA to analyze the data of our two-factorial design (2 genotypes x 3 flooding frequencies). Normal distribution of residuals were assessed visually prior to ANOVA testing. Due to the fully balanced study design, potential moderate deviations from homogeneity of variance between groups were considered unimportant for both ANOVA and PERMANOVA testing (Box 1954; McGuinness 2002; Anderson 2017). Along with ANOVA tests, we used Cochran's C test with $\alpha$ = 0.01 to test for single large variances (sensu McGuinnes 2002). When Cochran's test remained significant after log-transformation of data, nonparametric Kruskal-Wallis tests were conducted instead of ANOVA, which was only the case for a single parameter (decomposition rate, k). Beta dispersion tests conducted along with PERMANOVA indicated no significant heterogeneity of variances."*

References:
Box, G. E. (1954). Some theorems on quadratic forms applied in the study of analysis of variance problems, I. Effect of inequality of variance in the one-way classification. *The annals of mathematical statistics*, 290-302.
McGuinness, K. A. (2002). Of rowing boats, ocean liners and tests of the ANOVA homogeneity of variance assumption. *Austral Ecology*, 27(6), 681-688.
Anderson, M. J. (2017). Permutational multivariate analysis of variance (PERMANOVA). *Wiley statsref: statistics reference online*, 1-15. https://doi.org/10.1002/9781118445112.stat07841

Fig. 3: I find this figure to be really confusing, and it isn't clear to me what it adds that cannot already be gleaned from Figure 2. It isn't exactly clear what is being compared - did the authors pick whichever max-min was largest from among the 4 enzymes or is this averaged across enzymes and averaged across flooding treatments? Perhaps it is because I wasn't entirely clear on what the figure was showing, but it made it difficult for me to figure out what it added to the story. I would think that the authors could remove it all together, or if they choose to keep it, it would be helpful to have more clarity around how they aggregated their data and what conclusion can be drawn (e.g. is it average EEA change across all flooding treatments, sum of all EEA changes?).

We believe Figure 3 is a valuable illustration of the greater flooding effect on EEA in soils planted with the intolerant (vs. tolerant genotype) across all four enzymes assayed. We argue that this illustration and the associated statistics are important to demonstrate that the greater flooding effect on EEA in the intolerant genotype is a general phenomenon across all four enzymes despite the lack of significant effects in some enzymes when investigated separately (compare Table 1). In accordance with the reviewer's suggestion we will provide a more detailed description for the figure:

*"Figure 3: Maximum change in exo-enzyme activity (EEA) induced by the flooding treatment in soils planted with flooding tolerant vs. intolerant genotypes of Elymus athericus. EEA*

*change (%) refers to the difference between max and min average EEA of the three flooding treatments determined for each of the n = 4 exo-enzymes assayed (compare Table 1). Values are means and SE."*

Line 260:  One of the assumptions of permanova is that variances are equivalent. This appears to violate that assumption and should be tested for using beta dispersion tests. This would support your assertion that the variability is higher in the unadapted genotypes, but it would mean the authors would need to rethink their permanova.

We only partly agree with this statement: In contrast to both ANOSIM and Mantel test, PERMANOVA is very robust to heterogeneity for balanced designs. Furthermore, PERMANOVA is insensitive to differences in the correlation structure among groups (Anderson 2017). We agree, however, with the reviewer that the statement about variance differences should be supported using a besta dispersion test. Beta dispersion test indicated no significant difference between genotypes (p = 0.2). We therefore remove this statement from the revised version of the ms, while keeping the PERMANOVA results and the key conclusions derived from them.

References:
Anderson, M. J. (2017). Permutational multivariate analysis of variance (PERMANOVA). *Wiley statsref: statistics reference online*, 1-15. https://doi.org/10.1002/9781118445112.stat07841

Line 266: the authors should be commended for not conflating the taxonomic composition of their microbial communities with what might occur in situ.

We greatly appreciate the reviewers' support with this decision.

Discussion (first paragraph, principally) - I think the authors should keep the focus of their work on flooding, rather than climate change more broadly, as flooding was the only facet of climate change that was directly tested in this study.

We agree with the reviewer's suggestion. As stated above, we will carefully rephrase this section to avoid over-interpretation of our data.

**Anonymous Referee #2**

The authors present a fascinating study that reveals compelling evidence of plant genotype mediating the influence of inundation on coastal marsh soil microbial communities. Overall, the work makes a strong contribution that substantially advances understanding of whether (and how) intraspecific variation in ecologically dominant plants influences key aspects of carbon cycling in coastal marshes, which disproportionately influence the global carbon budget. Accordingly, the work warrants publication as it will be of great interest to a broad audience, ranging from evolutionary biologists to soil biogeochemists. Before publication, the authors should make some relatively minor but important revisions that will strengthen the presentation of their work, as follows:

- Most importantly, the authors should not use terms such as 'adaptation', 'adaptive', etc. to describe functional differences in the two genotypes included in the study

(and corresponding descriptions of the significance of plant intraspecific variation). These terms have very specific technical connotations, and their use requires evidence that phenotypic variation (i.e., functional variation) is (1) heritable; that the trait in question (2) responds to natural selection; and that responses in some way (3) relate to reproduction (i.e., fitness). The work presented here and elsewhere (i.e., Reents et al. 2021 https://doi.org/10.5194/bg-18-403-2021) does not present sufficient evidence that variation in tolerance to inundation is adaptive. Accordingly, the authors should substitute 'tolerant' and 'intolerant' (i.e., to inundation) to describe the two genotypes included in the study. Associated terms used elsewhere should be removed or replaced by technically suitable substitutes (e.g., the term 'plant intraspecific adaptations' should be replaced by 'plant intraspecific variation').

- From a methodological perspective, the source of the soil used for the experiment is somewhat concerning. It appears that the soil was not taken from the origin of either of the plant genotypes. This discrepancy should be noted as a caveat in the Discussion, as plant-microbe associations (and effects, outcomes thereof, etc.) can reflect provenance.
- From a statistical perspective, the number of correlation tests that were conducted is somewhat concerning. Further detail is warranted regarding the soil microbial parameters and plant biomass parameters that were examined. Also, depending on the number of tests conducted, significance should have been adjusted to account for multiple testing.

We thank the reviewer for the thoughtful comments.

(1) We agree with the reviewer and will change adapted and undadapted to tolerant and intolerant, respectively. Intraspecific adaptation will be changed to intraspecific variation throughout the manuscript.

(2) We agree with the reviewer and will provide a section on methodological considerations in the discussion.

**"Methodological considerations**

We previously demonstrated realistic plant-productivity responses to variations in flooding frequency simulated by the tidal-tank facility at Hamburg University (Reents et al. 2021). Therefore, we argue that also the present investigation on plant-soil interactions can provide relevant mechanistic insight into flooding effects on tidal-wetland functioning. However, owing to the artificial nature of the simulated tidal-wetland system, absolute effect sizes reported here need to be considered with caution. For the same reason, we refrain from providing a detailed interpretation of changes in single microbial taxa. One important caveat in this context is the restriction of our study to a single soil type. Because plant-microbe interactions in the rhizosphere can reflect provenance (e.g. Lonardo et al. 2018), future investigations will need to assess the generality of our findings using different combinations of plant genotype and soil type, including the native home soils from the locations at which the plants are sampled. We furthermore recommend repeating this experiment *in situ*, e.g. in

the form of reciprocal transplantations, in order to improve the quantitative understanding of plant genotype-mediated sea-level effects on soil microbial functioning."

References:
Di Lonardo, D. P., Manrubia, M., De Boer, W., Zweers, H., Veen, G. F., & Van der Wal, A. (2018). Relationship between home-field advantage of litter decomposition and priming of soil organic matter. *Soil Biology and Biochemistry*, *126*, 49-56.

(3) In accordance with the reviewer's remark, we will apply column-wise Bonferroni corrections for the correlations presented in Table 2. More detail on the plant and soil parameters used in these correlation analyses will be added to the methods section and, in part, to the table notes. The important correlations between aboveground biomass and microbial N acquisition and Δstabilization remained significant after Bonferroni corrections.

**Table 2: Correlations between plant biomass parameters and soil microbial activity parameters using a Bonferroni correction for multiple comparisons (α = 0.05 / number of pairwise comparisons). Shown are Pearson correlation coefficients (r). Significant correlations are highlighted in bold font (p ≤ α).**

|  | Aboveground | | Belowground | | Total biomass | |
|---|---|---|---|---|---|---|
|  | r value | p value | r value | p value | r value | p value |
| C activity | 0.03 | 0.857 | -0.02 | 0.909 | 0.01 | 0.972 |
| N activity | **0.41** | **0.004** | 0.23 | 0.120 | 0.36 | 0.013 |
| ΔC activity | 0.06 | 0.707 | -0.01 | 0.920 | 0.02 | 0.878 |
| ΔN activity | 0.37 | 0.010 | 0.09 | 0.539 | 0.26 | 0.075 |
| Decomp. rate *(k)* | -0.11 | 0.460 | -0.06 | 0.675 | -0.10 | 0.512 |
| Stabilization *(S)* | -0.05 | 0.724 | -0.15 | 0.316 | -0.12 | 0.438 |
| Δ*k* | -0.26 | 0.079 | -0.36 | 0.014 | -0.35 | 0.015 |
| Δ*S* | **-0.41** | **0.004** | **-0.45** | **0.001** | **-0.49** | **0.000** |

*Notes:* C activity = sum of C-acquisition enzyme activities (ß-glucosidase + cellobiosidase); N activity = sum of N-acquisition enzyme activities (aminopeptidase + chitinase); Decomp. rate (*k*) = decomposition rate constant (*sensu* Keuskamp et al. 2013); Stabilization (*S*) = stabilization factor (*sensu* Keuskamp et al. 2013); Δ = activity values in relation to the unplanted control (i.e. percentage change of planted vs. unplanted mesocosms) reflecting plant effects independent of direct (i.e. non-plant mediated) flooding effects.

Below are notes that relate these comments to specific elements of the text as well as other notes intended to improve the presentation of the authors' work.

SPECIFIC COMMENTS

ABSTRACT

L13. Topic sentence of the paragraph doesn't align with the content of the paragraph. Revise. Possibly combine and abbreviate the first and second sentences in the paragraph to create a new, more representative topic sentence. This is a good catch. It will be changed as suggested.

INTRODUCTION

L33. Change sentence structure to: "…It is therefore crucial to study the direct effects of climate change on soil microbial communities and resulting changes in ecosystem functioning. It is also important to examine plant-mediated, indirect effects (Bardgett et al., 2008; Van der Putten et al., 2013). Prior work on a wide range of ecosystems indicates that changes…" Will be changed accordingly.

L38. Change to "…on ecosystem C as well as greenhouse-gas and nutrient dynamics…" Will be changed accordingly.

L40. Topic sentence of the paragraph doesn't align with the content of the paragraph. Revise. Possibly combine and abbreviate the first and second sentences in the paragraph to create a new, more representative topic sentence. This is a good catch. It will be changed as suggested.

L57. Change to "These ecosystems are among the most effective…" Will be changed accordingly.

L67. Change to "The prevailing notion is that decomposition rates are inversely related to flooding." Will be changed accordingly.

L69. Change to "have demonstrated" Will be changed accordingly.

L75. Change to "control of microbial C cycling in wetland soils by plant processes" Will be changed accordingly.

L76. Change to "yet largely overlooked" Will be changed accordingly.

L82. Change to "we hypothesized that" Will be changed accordingly.

L87. Change to "by intraspecific adaptive variation" Will be changed accordingly.

METHODS

L97. Change to "The experiment was conducted" Will be changed accordingly.

L104. Methods concern: one source of soil was used for the experiment. It also appears that the soil was not taken from the origin of either of the plant genotypes. This discrepancy should be noted as a caveat in the Discussion, as plant-microbe associations (and effects, etc.) can reflect provenance. As stated above, we will discuss this point in the methodological considerations section that will be added to the revised version of the ms.

L111. Move "(n = 4)" to L109. Place after "four unplanted mesocosms" Will be changed accordingly.

L115. Provide more description here detailing (1) the extent of functional differences between the two genotypes (i.e., how different is the adaptive from the non-adaptive genotype, with regard to flood tolerance?). Also, provide a brief explanation of what was done to determine that the differences are (1) genetically-based, and (2) adaptive. It appears that sufficient work was done to demonstrate that the differences reflect heritable variation, but there isn't evidence (yet) that the differences are adaptive. It is entirely possible that functional differences can reflect heritable, non-adaptive differentiation.

As stated above, we agree with the reviewer comment concerning the lack of evidence of adaptive genetic variation. Therefore, we will refer to *flooding tolerant* or *sensitive* instead of *adapted* genotype throughout the revised version of the ms. We will additionally provide more information on the functional differences between the genotypes in the methods section:

*"Plants were collected in April 2015 from Elymus athericus stands on the island Schiermonnikoog, the Netherlands, that have previously been demonstrated to be dominated by genetically distinct populations of Elymus, i.e. flooding tolerant genotypes from the low marsh and intolerant genotypes from the high marsh (Bockelmann et al., 2003; Reents et al., 2021). In their natural environments, intolerant genotypes are grey-blue in color and produce tall shoots in dense stands, whereas tolerant genotypes are light green, produce more ramets and grow in a patchier distribution (Bockelmann et al. 2003). Recent common-garden experiments could demonstrate that some phenotypic differences between the genotypes are heritable. These include leaf color, shoot mass and length, as well as rhizome and root production (Mueller et al. 2021; Reents et al. 2021)."*

L172. How different were technical replicates with regard to community composition? i.e., what were the Bray-Curtis dissimilarity measures of replicates relative to other comparisons? This could be noted in the text or illustrated in a supplemental figure.

We will include an additional NMDS plot in the supplement showing ID labels (1-18) and technical-replicate labels (_1 or _2):

[Figure]

L175. What "soil microbial parameters and plant biomass parameters" were examined? Provide more detail here. Also, depending on the number of tests conducted, significance might need to be adjusted to account for multiple testing.

*Additional information will be added to the sentence as follows:*

*" [...] Canonical Correspondence Analysis (CCA) to explore the relationships between soil microbial parameters (i.e. activity of ß-glucosidase, cellobiosidase, chitinase, and leucine-aminopeptidase, litter decomposition rate and litter stabilization factor ) and plant biomass parameters (i.e. aboveground and belowground biomass; taken from Reents et al., 2021)."*

RESULTS

L220. The resolution of Figure 3 could be increased- the text appears a bit blurry.

*Figure 3 will be modified accordingly.*

L230-237. While this analysis was intended to be descriptive, the number of tests conducted warrants that significance be adjusted for multiple comparisons (i.e., Bonferonni corrections).

L240. Table 2. Presentation of significant correlations should be adjusted to reflect corrections accounting for multiple comparisons

*In accordance with the reviewer's remarks (L230-237 + L240), we will use column-wise Bonferroni corrections for the correlations shown in Table 2. See detailed response comment and revised Table 2 posted above.*

L246. Correct typo. Should read "sensu Keuskamp et al., 2013))" *Will be changed accordingly.*

L256. Delete "the" before "microbial" *Will be changed accordingly.*

L262. Change to "soil microbial community structure" *Will be changed accordingly.*

L263. Delete "the" before "community" Will be changed accordingly.

L264. Change to "is shown" Will be changed accordingly.

L274-276. Long and complicated sentence structure. Trim it back by deleting "in ecosystems" Will be changed accordingly.

L286. Again, it might very well be that the functional differences among plant genotypes are not adaptive. Will address this issue throughout the manuscript.

DISCUSSION AND CONCLUSIONS

L288. Topic sentence of the paragraph doesn't align with the content of the paragraph. Revise. Possibly combine and abbreviate the first and second sentences in the paragraph to create a new, more representative topic sentence. This will be changed as suggested.

L298. How is genotype-induced variability in plant biomass different from genotype?

Good catch! There is obviously no difference. The statement will be corrected for the revised version.

L321. Change to "…proven to be a powerful tool for characterizing…" Will be changed accordingly.

L327. Delete "parts of the" Will be changed accordingly.

L329. Reference to statistically significant correlations may have to be amended to account for concerns about multiple testing. This will be done based on the corrected results shown in the revised Table 2 posted above.

L335. Change sentence structure to "oxygen-deficient soils, like those found in coastal marshes. This suggests that it might be the most important mechanism, but strong genotype effects…" Will be changed accordingly.

L338-339. To support this premise, it would be good for the authors to refer to work done by Bernick et al. (MEPS 601:1-14 (2018) - DOI: https://doi.org/10.3354/meps12689) that illustrates heritable variation in nutrient acquisition among genotypes of a dominant plant (Spartina alterniflora) that engineers coastal marsh ecosystem attributes. The reference will be added accordingly.

L353. As already noted, the use of the terms "adaptations" and "adaptive" are not well supported and should be replaced by terms like "variation" or simply not used. Will be changed accordingly.

L359. Delete "neither of which were part of the present investigation" Will be changed accordingly.